# The Role of Patient Position in the Surgical Treatment of Supracondylar Fractures of the Humerus: Comparison of Prone and Supine Position

**DOI:** 10.3390/medicina59020374

**Published:** 2023-02-15

**Authors:** Marco Sapienza, Gianluca Testa, Andrea Vescio, Flora Maria Chiara Panvini, Alessia Caldaci, Stefania Claudia Parisi, Vito Pavone, Federico Canavese

**Affiliations:** 1Department of General Surgery and Medical Surgical Specialties, Section of Orthopaedics and Traumatology, P.O. “Policlinico Gaspare Rodolico”, University of Catania, Via Santa Sofia 78, 95123 Catania, Italy; 2Department of Pediatric Orthopedic Surgery, Jeanne de Flandre Hospital, Lille University Centre, 59000 Lille, France

**Keywords:** supracondylar humerus fracture, prone position, supine position, children, outcome, complications

## Abstract

*Background and Objectives*: Supracondylar fractures of the humerus (SCHF) make up about one-third of pediatric fractures and are the most common elbow fractures in children. Reduction and fixation of SC fractures can be performed with the patient in the prone or supine position. However, the role of the patient’s position during surgery is still unclear. The purpose of this systematic review is to evaluate, based on data from the literature, the role of patient position during closed reduction and fixation of pediatric SCHFs. *Materials and Methods*: A systematic review of the current literature from 1951 to 2021 was conducted according to PRISMA guidelines. Articles were identified from 6 public databases. Articles were screened and abstracted by two investigators and the quality of included publications (*n* = 14) was assessed (MINORS criteria). Statistical analyses were performed using R studio 4.1.2. *Results*: The systematic literature review identified 114 articles, from which, according to inclusion and exclusion criteria, 14 studies were identified. A total of 741 children were treated in the prone position and 538 in the supine position. The results of the systematic review showed that there were no statistical differences between the two positions with regard to clinical, radiographic, and complication outcomes. *Conclusions:*: The functional and radiographic outcome of displaced SCHFs is generally excellent regardless of the position, prone or supine, in which the patient is positioned for surgery. The choice of how to position the patient depends on the habit and experience of the surgeon and anesthesiologist performing the surgery.

## 1. Introduction

Supracondylar fractures of the humerus (SCHF) make up about one-third of pediatric fractures and are the most common elbow fractures in children. These fractures typically occur in children between 2 and 10 years of age (the common ages are 5–7 years), involve the nondominant side, and have an even gender distribution [1,2,3,4,5,6].

According to the amount of displacement, treatment can be conservative or surgical. In case of displaced SCHF, closed reduction and percutaneous fixation with 2 or 3 Kirschner wires (K-wires) is the gold standard treatment [6].

Surgery is performed under general anesthesia with the patient in a prone or supine position; the most common and widely accepted patient position during surgery is supine, although some surgeons position the patient in a prone position [6,7,8].

With the patient in supine position (Figure 1), traction followed by the hyperflexion of the elbow over 90° is needed to reduce the fracture [9,10]. With such a technique, an injury of the ulnar nerve can occur during elbow hyperflexion as the nerve dislocates anteriorly and becomes more vulnerable, especially during medial K-wire insertion [11,12].

When the patient is positioned prone (Figure 2), elbow hyperflexion is not necessary because gravity promotes reduction and maintenance of reduction with less manipulation [6,8]. Stabilization of the fracture with a lateral K-wire avoids flexion of the elbow during medial K-wire insertion and reduces the risk of ulnar nerve injury, which is unstable during the bending movement of the elbow [10].

Several studies have reported the advantages and disadvantages of both types of patient position, but none have analyzed the impact of patient position on SCHF reduction and fixation maneuvers.

The aim of this systematic review is to evaluate, based on data from the literature, the role of patient position during the closed reduction and fixation of pediatric SCHFs.

## 2. Materials and Methods

### 2.1. Search Strategy

We conducted a structured search of the following 6 databases from 1951 to 2021: PubMed, Embase, OVID, Web of Science, Scopus, and Cochrane Library. The following key words were primarily used in the literature search: “Children” OR “childhood” OR “pediatric” and “supracondylar humerus fracture” OR “distal humerus” OR “Gartland” and “pinning” OR “surgery” and “position” OR “supine” OR “prone”. We used Boolean operators to combine topic words with keywords and search for references in related literatures. Our search adhered to the Preferred Reporting Items for System Review and Meta-analysis (PRISMA) guidelines for a systematic review of rates [13].

### 2.2. Inclusion and Exclusion Criteria

Inclusion criteria were as follows: (1) studies of any level of evidence; (2) studies written in English; (3) studies of patients aged ≤18 years at the time of treatment and studies from which position of the patient (prone or supine) could be extracted; (4) studies reporting clinical or preclinical results; (5) text of full article available; (6) studies with a methodological index for non-randomized studies (MINORS) quality evaluation score >14 points (Table 1) [14].

Exclusion criteria were as follows: (1) review articles, case reports, and articles written in languages other than English; (2) studies from which the position of the patient could not be extracted; (3) articles dealing with other elbow injuries; (4) studies with a methodological index for non-randomized studies (MINORS) quality evaluation score ≤13 points [14].

### 2.3. Quality Evaluation Using the MINORS Checklist

The quality of included non-comparative studies was assessed using the MINORS item quality evaluation checklist [14] based on the following 8 indicators: a clearly stated aim; inclusion of consecutive patients; prospective data collection; endpoints appropriate to the aim of the study; unbiased assessment of the study endpoint; follow-up period appropriate to the aim of the study; loss to follow-up <5%; prospective calculation of the study size. The items are scored 0 (not reported), 1 (reported but inadequate), or 2 (reported and adequate).

Two researchers (SM and VA) scored the literatures independently according to the MINORS criteria checklist. In the event of a conflicting evaluation of an article, after discussion, the two evaluators proposed a common score.

### 2.4. Data Extraction

Two researchers (MS and VA) independently screened the titles and abstracts and then reviewed the full text of eligible articles; discrepancies were resolved by the third researcher (VP); the PRISMA flowchart for the selection and screening method is provided in Figure 3.

The following data were extracted for analysis: the characteristics of the study, including the first author, year of publication, country of origin, number of patients, number of SCHF, and the clinical characteristics of patients, including sex, age at onset, age at operation, patient position during surgery (prone or supine), follow-up years, complications, and final outcome.

### 2.5. Statistical Analysis

Statistical analyses were performed using R studio 4.1.2 (RStudio Inc., Boston, MA, USA).

## 3. Results

### 3.1. Search Results

Our search strategy identified 114 articles, of which 72 were excluded based on the title and abstract, leaving 42 eligible articles. Eleven of the forty-two articles were excluded because the full text could not be found, three were excluded because the topic was not relevant to the current study or the patient position (supine or prone) was not mentioned or could not be extracted, and fifteen because they were written in language other than English; thirteen articles were identified based on inclusion and exclusion criteria (Figure 3).

Overall, 1279 patients (65% boy and 35% girls) were included. Mean age at treatment was 6.7 years (range, 5.2–9.5), and mean follow up time was 26 months (range, 3–81); 80% of SCHFs were Gartland 3 (*n* = 1023), and 20% were Gartland 2 (*n* = 256). The percentage of patients with displaced SCHF treated in prone are 741 (57.9%) and in supine position are 538 (42.1%).

### 3.2. Basic Characteristics of Included Studies

Table 2 provides a summary of all included studies [2,7,9,15,16,17,18,19,20,21,22,23,24].

### 3.3. Surgical Treatment

Of the fourteen studies, 10 retrospective, 2 prospective, and 2 cohort studies that assessed surgical treatment of SCHF with the patient in supine or prone position were identified.

Three studies evaluated the outcome in patients treated in supine position, five in those treated in prone position, and six compared patients treated in prone or supine position.

### 3.4. Effect of Prone or Supine Position on Clinical and Radiographic Outcome

After closed reduction treatment, the results were judged to be excellent according to Flynn’s criteria [26] while radiographic results were satisfactory considering Baumann’s and humero-capitellar angles [27].

### 3.5. Effect of Prone or Supine Position on Surgical Outcome

Among patients operated in the supine position, 15 out of 538 required open reduction (2.8%) while among those operated in the prone position, 4 out of 741 required open reduction (0.53%). In all cases, these were Gartland 3 fractures. As a matter of fact, open reduction as second chance, neither as first option treatment is always analyzed [2,9,15,21]. In two cases where closed reduction is not achieved in the prone position; both of them were remanipulated into the supine position [7].

Among patients operated in the supine position (538), there were 31 cases of ulnar nerve paresthesia (6%), 4 cases of anterior interosseous nerve injury (0.8%), 3 cases of posterior interosseous nerve injury (0.6%), and one case of radial nerve injury (0.2%). In addition, in the supine patient group, the following vascular injuries were identified: 1 injury of the brachial artery (0.2%), capillary pulse was poorly detectable with pulse oximeter at 24 h after surgery in 5 cases (1%), and reduced capillary refill by >50% in 1 case (0.2%). Pin tract infection was recorded in 32 cases (6%); 6 cases of poor ROM were detected (1.1%).

During follow-up, 4 patients developed cubitus varus (1.1%) (Table 3).

Among patients operated in the prone position (741), there were 6 cases of paresthesia (0.8%) [11], one case of compartment syndrome with median nerve palsy (0.15%), and no ulnar nerve injury (9); 8 cases of poor ROM were detected (1%). No vascular injury was reported while capillary pulse was difficult to record with pulse oximeter at 24 h after surgery in 4 cases (0.6%), and capillary refill time was reduced by >50% in 2 patients (0.3%) [16].

During follow-up, 2 patients developed cubitus varus (0.3%), 2 patients had floating elbow (0.3%), and 1 patient developed mild hyperextension (0.15%); in addition, 2 fractures lost reduction (0.3%), and 6 pin tract infections (0.8%) were recorded (Table 3).

No differences were found between the two groups with respect to radiation exposure, pin placement errors, and duration of surgery.

### 3.6. Anaesthesia Time

Only one article compared the supine and prone position with respect to anesthesia time, which was reported to be higher in patients treated prone (46.7 ± 7.6 vs. 37.2 ± 5.9 min; *p* < 0.001) [13].

## 4. Discussion

Nowadays, the literature showed that most surgeons prefer the supine position for the surgical treatment of SCHFs [2,8,28]. Our results also show that once optimal fracture reduction is achieved, the functional and radiographic outcome is generally excellent regardless of the position (prone or supine) in which the patient is placed for surgery.

Our analysis found that the complication rate when considering the patient’s position, prone or supine, is similar. Likewise, the open reduction rate in the two groups of patients is comparable, 0.53% in the prone position and 2.8% in the supine position (*p* > 0.05).

According to the literature, a relatively frequent complication that can happen with the patient in the supine position (Figure 1) is an ulnar nerve injury, mostly during medial K-wire insertion when the anterior displacement of the ulnar nerve occurs during the elbow hyperflexion reduction maneuver, which makes the nerve more vulnerable and more susceptible to iatrogenic injury [9,10]. Basant et al. (2012) reported excellent functional and aesthetic outcome according to Flynn’s criteria, although ulnar nerve injury was found in 26 cases, 18 of which were detected in the immediate postoperative period; the authors also found 32 cases of pin tract infection [25]. In addition, Herzog et al. (2013) reported 8 cases (15%) of ulnar nerve injury that resolved without persistent deficit and one case of brachial artery injury (1.88%); the authors pointed out that the supine position allows K-wire insertion more distally, thus allowing better control of the fragments [24]. Turgut et al. (2014) reported that closed reduction and percutaneous fixation of SHCFs with patients in the supine has a lower complication rate and that it should be preferred to open reduction. However, in a case where open reduction is needed (15 out of 538 cases in our review), the anterior, anteromedial, or anterolateral approach can be performed, especially in the case of neurovascular injury, open fractures, and compartment syndrome [29].

Fowler et al. (2006) was the first author to advocate and report excellent results with closed reduction and percutaneous fixation of SCHFs in prone patients. In most cases, patients did not suffer iatrogenic injury to the ulnar nerve, and no fractures lost reduction. There was no infection of the K-wire tract or loosening of the K-wire that necessitated its early removal. Fowler et al. emphasized the advantage of the prone position, in which hyperflexion of the elbow is not needed to reduce the fracture because gravity favors both the reduction and its maintenance [6,15], and it is not essential for medial K-wire insertion, which does not expose the ulnar nerve to iatrogenic injury. Indeed, in their study, Fowler et al. report no iatrogenic injury to the ulnar nerve [23]. Havlas et al. (2008) and De Pellegrin et al. (2008) confirmed that the prone position allows safe percutaneous positioning of the medial K-wire and did not record any iatrogenic ulnar nerve injury. In addition, the prone position with the forearm suspended on the arm board takes advantage of gravity to facilitate fracture reduction. Havlas et al. and De Pellegrin et al. reported a good to excellent outcome in all patients according to Flynn’s criteria [15,28]; similar findings were also reported by Kao et al. (2006, 2008, and 2014) [16,17,23].

Similarly, Kao et al. (2008) reported 97% (*n* = 9) of excellent clinical outcome according Flynn’s criteria, as well as complete fracture union in all cases with the patients treated in the prone position with complete union. Similar results were confirmed in another work by Kao et al. (2007), although the number of patients was relatively low (*n* = 10) [16,17].

However, if open reduction is necessary in the patients treated in the prone position (4 out of 741 cases in our review), the posterior approach is possible without changing the patient’s position; if an anterior, anteromedial, or anterolateral approach is needed, the patient must be positioned prone, thus changing the patient’s position [7,18]. An additional advantage of the prone position is the use of the C-arm in two standard planes; therefore, it is the C-arm that rotates 90° for the lateral view and not the patient’s elbow, in contrast to the supine position. Recently, even with the patient in the supine position, moving the C-arm for the lateral view has been considered; the visualization of the correct position of the K-wire after reduction is difficult because of the overlap of the forearm and distal humerus in the hyperflexed elbow [9,10].

Among the studies included in our review, three papers (Pescatori et al., Bãlãnescu et al. and Pavone et al.) carried out a comparative analysis (level III of evidence) between the supine and prone position of children operated for a SCHF.

Pescatori et al. (2012) reviewed 68 patients (34 treated in supine and 34 in prone position) and reported (*n* = 2) 2 cases of *cubitus varus* (5.88%) in the supine and none in the prone group, though the difference was not statistically significant (*p* = 0.165); all fractures achieved complete bone healing, and no cases of infection were recorded. At the last follow-up visit, the range of movement was comparable to the contralateral side in all patients, irrespective of the position during surgery. Similar results were reported by Bãlãnescu et al. (2013) [19,20].

Pavone et al. (2020) compared the clinical and radiographic outcome of patients with displaced SCHF treated in the supine (*n* = 34) and prone (*n* = 25) position. According to Flynn’s criteria, patients in the supine group had an excellent functional and cosmetic outcome in 94.1% of cases (mean Mayo Elbow Performance Score: 96 ± 3.8) and 97% of cases, respectively; in patients positioned prone, the percentage were 92% (mean Mayo Elbow Performance Score: 97.8 ± 3.3) and 100%, respectively. Radiographically, the mean difference of Baumann’s angle between the injured and the uninjured limb was 5.5° ± 1° in the supine position group and 5.1° ± 1.1° in the prone position group. The supine group had 0.8 cm asymmetry in 2.9% of cases, ulnar nerve paresthesia in 5.9% of cases, which resolved spontaneously over a period of 2 to 3 months. No cases of ulnar nerve paresthesia were recorded in the prone position group, though 2.9% of patients developed mild *cubitus varus*. Pavone et al. concluded that both positions are comparable in terms of functional, cosmetic, and radiographic outcome [2].

As far as the length of anesthesia is concerned, two papers (Guler et al. and Venkatadass et al.) considered this parameter. Guler et al. (2016) reported that the length of anesthesia was shorter when surgery was performed with the patient in supine compared to prone position; all other variables including age, sex, fracture type, side, surgical time, fluoroscopy time, time from trauma to surgery, number of reduction attempts, number of pinning attempts, hospitalization, follow-up, outcome Flynn’s criteria, Baumann’s angle, and lateral radio-capitaller angle were comparable [21]. In contrast, Venkatadass et al. (2015) reported that the anesthesia risk was comparable irrespective of the position of the patient during surgery [7].

## 5. Conclusions

The functional and radiographic outcome of displaced SCHFs is generally excellent regardless of the position, prone or supine, in which the patient is positioned for surgery. The choice of how to position the patient depends on the habit and experience of the surgeon and anesthesiologist performing the surgery. However, in recent years, the literature has highlighted the potential benefits, in terms of clinical and radiographic outcomes, of surgically treating the patient with an SCHF in the prone position.

## Figures and Tables

**Figure 1 medicina-59-00374-f001:**
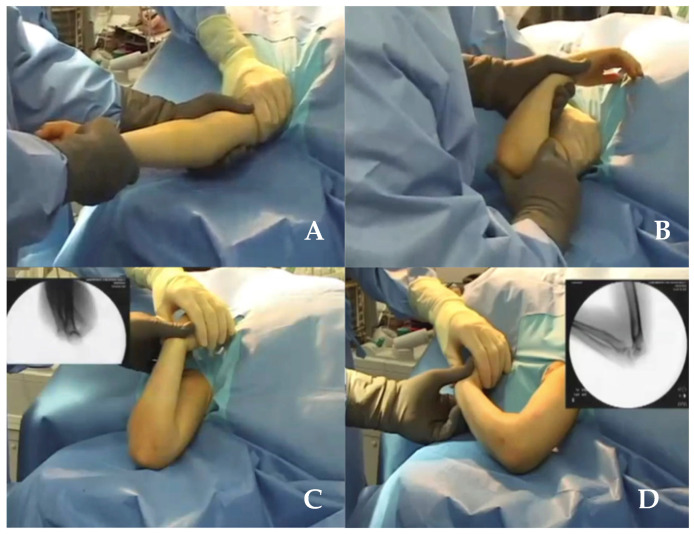
(**A**) Longitudinal traction; (**B**) Reduction manoeuvres and pinning in supine position; (**C**) Maximal hyperflexion in pronation; (**D**) Progressive flexion with pronation of the forearm.

**Figure 2 medicina-59-00374-f002:**
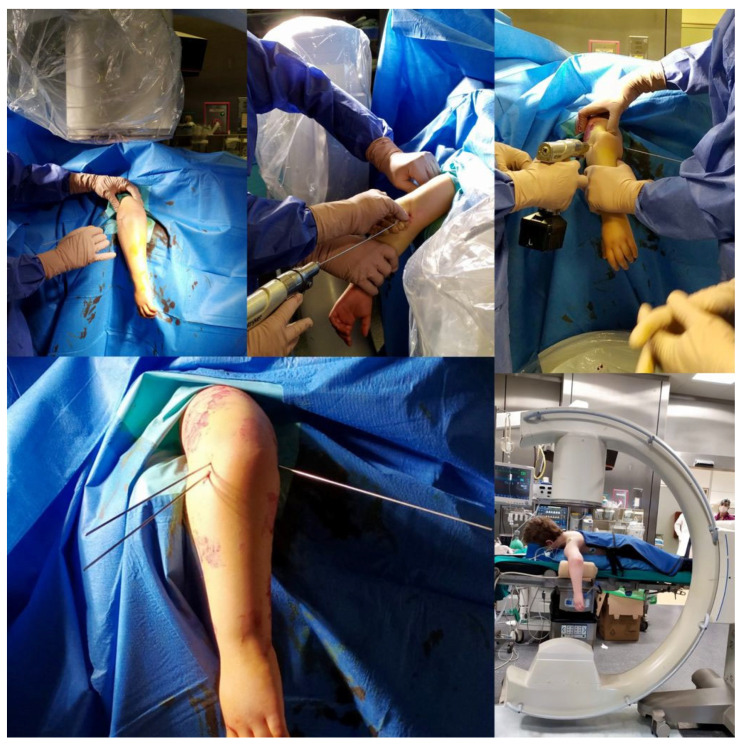
Pinning in prone position.

**Figure 3 medicina-59-00374-f003:**
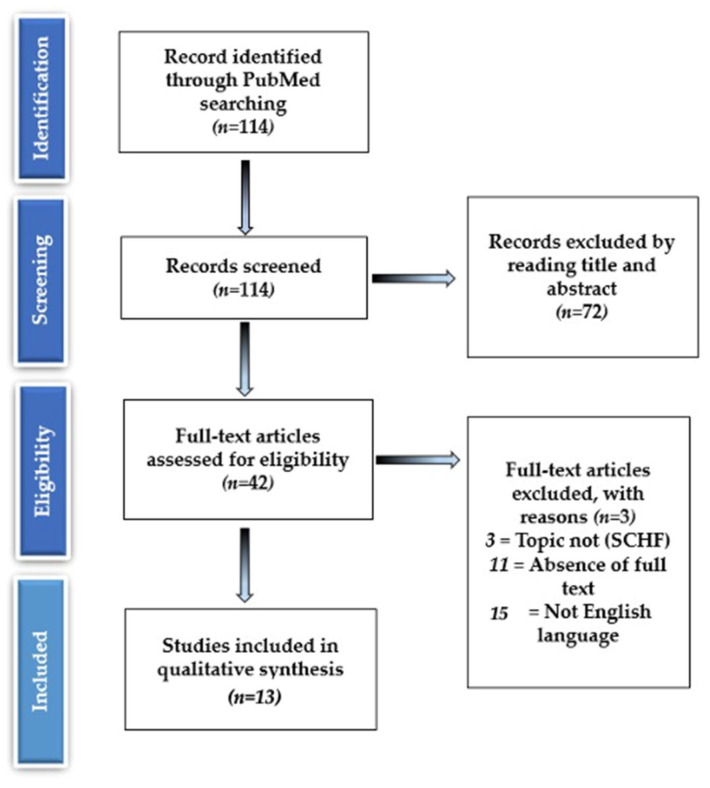
The PRISMA flowchart for the selection and screening method.

**Table 1 medicina-59-00374-t001:** The revised and validated version of MINORS.

Methodological Items for Non-Randomized Studies	Score ^†^
**A clearly stated aim:** the question addressed should be precise and relevant in the light of available literature **Inclusion of consecutive patients:** all patients potentially fit for inclusion (satisfying the criteria for inclusion) have been included in the study during the study period (no exclusion or details about the reasons for exclusion) **Prospective collection of data:** data were collected according to a protocol established before the beginning of the study **Endpoints appropriate to the aim of the study:** unambiguous explanation of the criteria used to evaluate the main outcome which should be in accordance with the question addressed by the study. Also. the endpoints should be assessed on an intention-to-treat basis. **Unbiased assessment of the study endpoint:** blind evaluation of objective endpoints and double-blind evaluation of subjective endpoints. Otherwise the reasons for not blinding should be stated **Follow-up period appropriate to the aim of the study:** the follow-up should be sufficiently long to allow the assessment of the main endpoint and possible adverse events **Loss to follow up less than 5%:** all patients should be included in the follow up. Otherwise. the proportion lost to follow up should not exceed the proportion experiencing the major endpoint **Prospective calculation of the study size:** information of the size of detectable difference of interest with a calculation of 95% confidence interval, according to the expected incidence of the outcome event, and information about the level for statistical significance and estimates of power when comparing the outcomes *Additional criteria in the case of comparative study* 9. **An adequate control group:** having a gold standard diagnostic test or therapeutic intervention recognized as the optimal intervention according to the available published data 10. **Contemporary groups:** control and studied group should be managed during the same time period (no historical comparison) 11. **Baseline equivalence of groups:** the groups should be similar regarding the criteria other than the studied endpoints. Absence of confounding factors that could bias the interpretation of the results 12. **Adequate statistical analyses:** whether the statistics were in accordance with the type of study with calculation of confidence intervals or relative risk	

^†^ The items are scored 0 (not reported), 1 (reported but inadequate) or 2 (reported and adequate). The global ideal score being 16 for non-comparative studies and 24 for comparative studies.

**Table 2 medicina-59-00374-t002:** Results of selected studies.

Author	Type of Study	Number of Patients (Sex Ratio):	Age (Years)	Follow-Up (Months)	Position	Type of Fracture According to Gartland Classification	Clinical Evaluation and Results	Radiographic Evaluation and Results	Open Reduction Required	Complications	Limits
Vojtech Havlas et al. [15] (2008)	Rtc	455 (261 M/194 F)	7.5	3 ± 6	Prone	II–III			Loss of reduction: 3.5% (*n* = 16). Not analyzed the second procedure	Loss of reduction: 3.5% (*n* = 16) Pin tract infections 1.3% (*n* = 6).	Retrospective study. Short follow-up.
Hsuan-Kai Kao et al. [16] (2014)	Rtc	34 (22 M/12 F)	5.2	17.4	Prone	III	Flynn’s criteria: excellent 31, good 2, poor 1	Baumann’s angle 5.1 6 3.9°. Humerocapitellar angle 9 6 10° (range 0–55°)	0 out of 34	Loss of reduction: 5.8% (*n* = 2)	Retrospective case series study no comparative radiographs of the noninjured elbow no control group treated with other procedures
K. Venkatadass et al. [7] (2015)	Ct	26 (20 M/6 F)	6.8	12	Prone	III	Flynn’s Criteria (Cosmetical Factors: Excellent 14/Good 4/Fair 2/Poor 1) Functional Factor (Excellent 9/Good 8/Fair 3/Poor3)	Baumann’s angle 18.46° (7 outliers in prone group)	0 out of 26. Closed reduction not achieved in 2/26 in prone position.	Compartment syndrome with median nerve palsy 3.84% (*n* = 1) Cubitus varus 7% (*n* = 2) Poor ROM 10% (*n =* 3)	Small sample size
Hsuan-Kai Kao et al. [17] (2017)	Rtc	10 (7 M/3 F)	9.5	17.8	Prone	III	Flynn’s criteria excellent in 9 patients/poor in 1 patient	Baumann’s angle 3.5 + 1.9 (range: 0–7) Humerocapitellar angle 7.9 + 7.4 (range: 1–24)	2/10 caused by ulnar preoperatory deficiency		Small sample size Retrospective case series no comparative radiographs of the noninjured elbow no control group
R. Bãlãnescu et al. [19] (2013)	prospective	40		3	Supine	III	Clinical evaluation after 1 year: elbow flexion +130/140 degrees; elbow extension 0/+ 5 degrees	Radiographic evaluation after 1 year postoperatively: excellent or very good results in 38 patients (95%)	0/40	24 h after surgery: The capillary pulse was found difficult to record using a pulse oximeter in 5 (12.5%) patients (meaning that it displayed values varying >20%) Capillary refill time reduced by >50% in 1 patient (0.02%) Paraesthesia in 4 patients (10%) Finger function affected in 1 patient. 3 months postoperatively: elbow flexion <90 degrees in 3 patients (7%)	Retrospective study Small sample size, no multicentric study. No long-term follow-up.
Ezio Pescatori et al. [20] (2012)	Rtc	34 (19 M/15 F)	6.1	46	Supine	III	Flynn’s criteria (Functional: 32 excellent, 2 good, 0 acceptable) (Cosmetic: Humerus–ulnar–carpal angle 30 excellent, 1 good,1 acceptable)		1 out of 34	Cubitus varus 5.88% (*n =* 2)	Retrospective study Small sample size.
Olcay Guler et al. [21] (2016)	Rtc	27 (15 M/12 F)	6.9 ± 1.5	22.8 ± 9.9	Prone	III	Flynn’s criteria (very good: good 23:4)	Baumann’s angle 73.1° ± 3.5°. Lateral radiocapitaller angle 41.7° ± 4.2°.	-		Retrospective study Small sample size.
Ali Turgut et al. [22] (2014)	Rtc	19	(Mean age: 7 y and 6 m, range: 3 y and 2 m to 12 y and 10 m)	13.3	Supine	III	Elbow flexion-extension range of motion 146° (range: 130° to 160°). Carrying angle decreased in 7 elbows, increased in 6 and remained the same in 6.		0 out of 19		Retrospective study Small sample size.
M. De Pellegrin et al. [9] (2008)	Rtc	45 (32 M/13 F)	6.5		Prone	III	Flynn’s criteria (Excellent 44/good 1) Cubital angle (44 excellent/1 deviation varus of 6° and hyperextension of 10° (good result)		-		Retrospective study Small sample size.
T. P. Fowler et al. [23] (2006)	Rtc	19		until fracture healing	Prone	(14 Gartland type III/5 Gartland type II)			2 out of 19	Ulnar nerve hypermobility with a propensity to displace anteriorly with elbow flexion >50% of children aged 6 to 10 years	Retrospective study Small sample size.
Vito Pavone et al. [2] (2020)	Rtc	25	5.9 ± 2.3	59.9 ± 12.8	Prone	III	Flynn’s criteria excellent cosmetic outcome in 23 subjects (92.0%) and good in 2 (8.0%). Functional factor was satisfactory in 100% of patients. MEPS 97.8 ± 3.3 (range 91–100) Final follow-up, the range of motion gave a flexion range of about 113.6° ± 11.2° (range 94°–137°), extension of 2.9° ± 2.2° (range 0°–10°), and supination to pronation of about 86.2° ± 2.2° (range 84°–90°).	Baumann’s angle 5.1° ± 1.1° (range 2.8°–6.6°)	-	Mild hyperextension 4% (*n* = 1) Local infection 8% (*n* = 2)	Retrospective study Small sample size. Lack of objective measurements.
Herzog M. et al. [24] (2013)	Rtc	106 (30 M/23 F)	6.1	1	Supine	II–III	ROM	Baumann’s angle. Humeral capitellar angle.	8 out of 106	Vascular injury 1.88% (*n* = 1) Nerve injury 15% (*n* = 8)	Retrospective study Small sample size. Lack of objective measurements.
Basant Kumar Bhuyan M.S [25] (2012)	Prospective	277	6	54	Supine	II–III	Flynn’s criteria (excellent 202/good 68/fair 5, poor 2). Carrying angle 10.65°.	Baumann’s angle. Humeral capitellar angle.	6 out of 257	Pin tract infection 11.55% (*n* = 32) Ulnar nerve injury 9.38% (*n* = 26)	Small sample size no multicentric study.

M = male; F = Female; Rtc = randomized controlled trial, Ct = controlled trial; ROM = range of motion.

**Table 3 medicina-59-00374-t003:** Comparison between the complications reported in prone vs. supine position (randomized controlled trials and controlled trials only).

PRONE	SUPINE
Local infection 0.8%(*n* = 2)	Local infection 6% (*n* = 32)
Cubitus varus 0.3% (*n* = 2)	Cubitus varus 1.1% (*n* = 2)
Poor ROM 1% (*n* = 8) Paraesthesia 0.8% (*n* = 6)	Poor ROM 1.1% (*n* = 6) Vascular injury 1.88% (*n* = 1) Nerve injury 7.6% (*n* = 39)
Compartment syndrome with median nerve palsy 0.15% (*n* = 1)	

## Data Availability

Data available on request due to restrictions (privacy and ethical).

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
