# Peer review of "The Role of Patient Position in the Surgical Treatment of Supracondylar Fractures of the Humerus: Comparison of Prone and Supine Position"

_medicina, 2023, doi:10.3390/medicina59020374_

Round 1

Reviewer 1 Report

 Title:

Ok.

Abstract:

See my comments in result section.

Introduction:

 Although there are case reports and of course the fractures can occur in children aged 2-10, the common ages are 5-7, I suggest editing this line or adding this information.

Stabilization of the fracture with a lateral K-wire avoids flexion of the elbow during medial K-wire insertion and reduces the risk of ulnar nerve injury.

This sentence is unclear, please clarify.

 Methods:

 There is no reference regarding inclusion or exclusion depending on the follow-up time

In my opinion, this is a fundamental issue, as can also be seen in the results and studies that were finally included, some of which have a follow-up time of only up to 3 months. I'm not sure if broad conclusions can be drawn based on the studies that were finally included.

 Results:

“65% Men and 45% women”

Boys and Girls would be more suitable.

 “Mean age at treatment was 6.7 years (range, 5.2-9.5)”

See my previous comment regarding age of the patients. This result emphasis the previous comment.

 Table 2 – add serieal numbers to the studies

In this table there are 20 studies. And the authors stated that 14 studies were included. Please clarify.

 Please clarify why the study by Hsuan-Kai Kao et al. with only 10 patients were included in this systemic review, it is a very small case series,

In my opinion, including this study along with the other studies slightly detracts from the value of the review article.

 Please clarify if all surgical treatments were CRIF or any of the study included cases treated via ORIF as this can alter the results and conclusions.

 No differences were found between the two groups with respect to radiation expo-27 sure, pin placement errors and duration of surgery.”

 In the abstract the authors stated that “Anaesthesia time was higher in patients treated prone compared to those treated in supine position (46.7±7.6 vs. 37.2±5.9 minutes; p<0.001).” please clarify.

 Table 3 is not displayed clearly, the general number of patients in each of the groups must be added, percentages must be entered, only in one parameter the percentage is specified now.

 Only one article compared supine and prone position with respect to anaesthesia  time, which was reported to be higher in patients treated prone (46.7±7.6 vs. 37.2±5.9 33 minutes; p<0.001). [13] "

Based on one study, the authors stated in the abstract the same result.

This is misleading, because according to the abstract it seems that this is a data analysis of the review article and it is not detailed that the results are  based on only one article.

 Discussion:

“Our study showed that most surgeons prefer the supine position for the surgical 36 treatment of SCHFs.”

Based on Table 2 , 11\20 studies used the prone position

Please clarify.

 Line 43-47 is a repetition of the introduction.

 “If open reduction is required, the surgical approach can be anterior, anteromedial, or 62 anterolateral and it is not possible or is more difficult when the patient is in the prone 63 position.”

See my previous comment. If studies with supine position included ORIF this alter the entire systematic review.

 Line 65-68 is a repetition of the introduction.

 The discussion contains a great deal of data on studies carried out in the past, and it is difficult to follow and understand the purpose of the entire collection of paragraphs.

I suggest conducting the discussion and focusing on the main results and trying to extract common information and similar conclusions raised from the review.

Author Response

Thanks for your support and suggestions, in the text you will find the underlined changes

Introduction:

I have corrected and processed the correct sentences by you.

Methods:

Thank you for your important comment. In Table 2, the limits of individual studies are specified.

Results:

Thank you for your suggestions, mistakenly I had inserted an outdated table with duplicate articles.

Unfortunately, the study of Hsuan-Kai Kao et al had a small sample size of patients, but I specified clearly within the limits of the study.

All were treated at first with CRIF technique.

“No differences were found between the two groups with respect to radiation expo-27 sure, pin placement errors and duration of surgery.”

 In the abstract the authors stated that “Anaesthesia time was higher in patients treated prone compared to those treated in supine position (46.7±7.6 vs. 37.2±5.9 minutes; p<0.001).”

We talk about anesthesiological times, not surgical times.

I have inserted the percentages in the table and arranged the reference in the text

“Anaesthesia time was higher in patients treated prone compared to those treated in supine position (46.7±7.6 vs. 37.2±5.9 minutes; p<0.001).”

I deleted this sentence from the abstract results.

Discussion:

Thank you for your comments,

I corrected the first sentence and removed the repetition of the introduction.

I just talked about open reduction for completeness of information in the discussion.

I have removed the repetition of the introduction and somewhat eased the information in the discussion.

Kind regards,

Marco Sapienza.

Reviewer 2 Report

The author made a statistical study about Supracondylar Fractures. The research is interesting however the manuscript should improved.

 The authors should change the form to made the citations for instance the authors put [1,2,3,4,5,6], they should change to [1-6]. Also page 5 table 2.

 In Figure 1 the authors should identify clearly every imbibed picture

 Statistically what kind of Supracondylar Fracture was the more recurrent?

 Statistically what kind fixation instruments  (plate, nails, screw) were more used?

Author Response

Thanks for your support and suggestions, in the text you will find the underlined changes

I changed the form of the citations.

I have clearly indicated each image in figure 1 as required.

Unfortunately, it was not present in every study analyzed the most frequent type of fracture so we decided not to specify it.

The main means of synthesis used are K-wires

Kind regards,

Marco Sapienza

Round 2

Reviewer 1 Report

"(65% boy and 45% women)" please correct

Please clarify  - in the text (!) if all surgical treatments were CRIF or any of the study included cases treated via ORIF as this can alter the results and conclusions.

The discussion contains a great deal of data on studies carried out in the past, and it is difficult to follow and understand the purpose of the entire collection of paragraphs.

I suggest conducting the discussion and focusing on the main results and trying to extract common information and similar conclusions raised from the review.

the author\s did not adressed this issuue. please adress and clarify.

Author Response

Thank you for your valuable suggestions to improve our systematic review.

We have inserted in Table 2 and in the text the information about the OPEN treatment.

We have modified the discussion following your indications and focusing on the results, I hope that now everything is clearer.

Kind regards.